# Learning Discrete Representation with Optimal Transport Quantized Autoencoders

## Abstract

Vector quantized variational autoencoder (VQ-VAE) has recently emerged as a powerful generative model for learning discrete representations. Like other vector quantization methods, one key challenge of training VQ-VAE comes from the codebook collapse, *i.e.* only a fraction of codes are used, limiting its reconstruction qualities. To this end, VQ-VAE often leverages some carefully designed heuristics during the training to use more codes. In this paper, we propose a simple yet effective approach to overcome this issue through optimal transport, which regularizes the quantization by explicitly assigning equal number of samples to each code. The proposed approach, named OT-VAE, enforces the full utilization of the codebook while not requiring any heuristics such as stop-gradient, exponential moving average, and codebook reset. We empirically validate our approach on three different data modalities: images, speech and 3D human motions. For all the modalities, OT-VAE shows better reconstruction with higher perplexity than other VQ-VAE variants on several datasets. In particular, OT-VAE achieves state-of-the-art results on the AIST++ dataset for 3D dance generation. Our code will be released upon publication.

## 1 Introduction

Unsupervised generative modeling aims at generating samples following the same distribution as the observed data. Recent deep generative models have shown impressive performance in generating various data modalities such as image, text and audio, owing to the use of a huge number of parameters in their models. The well known examples include VQ-GAN (Esser et al., 2021) for high-resolution image synthesis, DALLE (Ramesh et al., 2021) for realistic image generation from a description in natural language, and Jukebox (Dhariwal et al., 2020) for music generation. Surprisingly, all these models are based, at least partly, on Vector Quantized Variational Autoencoders (VQ-VAE) (Van Den Oord et al., 2017). The success of VQ-VAE should be mostly attributed to its ability of learning discrete, rather than continuous, latent representations and its decoupling of learning the discrete representation and the prior. The quality of the discrete representation is essential to the quality of the generation and our work improves upon the discrete representation learning for arbitrary data modality.

VQ-VAE is a variant of VAEs (Kingma & Welling, 2014) that first encodes the input data to a discrete variable in a latent space, and then decodes the latent variable to a sample of the input space. The discrete representation of the latent variable is enabled by vector quantization, generally through a nearest neighbor look up in a learnable codebook. A new sample is then generated by decoding a discrete latent variable sampled from an approximate prior, which is learned on the space of the encoded discrete latent variables in a decoupled fashion using any autoregressive model (Van Den Oord et al., 2017). Despite its promising results in many tasks of generating complex data modalities, the naive training scheme of VQ-VAE used in (Van Den Oord et al., 2017) often suffers from codebook collapse (Takida et al., 2022), *i.e.* only a fraction of codes are effectively used, which largely limits the quality of the discrete latent representations. To this end, many techniques and variants have been proposed, such as stop-gradient along with the commitment and embedding loss (Van Den Oord et al., 2017), exponential moving average (EMA) for codebook update (Van Den Oord et al., 2017), codebook reset (Williams et al., 2020) and a stochastic variant (SQ-VAE) (Takida et al., 2022).

Interestingly, the idea of vector quantization has also been explored in the related field of self-supervised learning, though it generally relies on unsupervised discriminative modeling by only obtaining data features that can be easily generalized to downstream tasks. The seminal work by Caron et al. (2018) used an encoder and a clustering algorithm to learn discriminative representations of the data. The clusering algorithm used in Caron et al. (2018), namely K-means, could be interpreted as an offline version of the vector quantization used in VQ-VAE. Similar to the codebook collapse, some clusters were observed to have a single element, known as cluster collapse. To address this problem, Asano et al. (2020) have proposed an optimal transport (OT) based clustering method to explicitly enforce the equipartition of the clusters. Caron et al. (2020) have later proposed an online version of their algorithm for dealing with large-scale datasets.

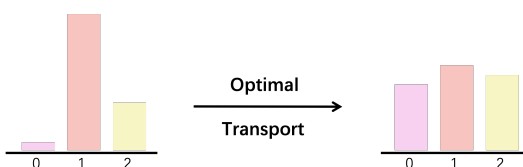

Figure 1: Optimal transport (OT) explicitly enforce the equipartition of the clusters.

In this work, we reformulate VQ-VAE under the framework of Wasserstein Autoencoders (WAE) (Tolstikhin et al., 2018), providing a natural connection between distribution matching in the latent space of VQ-VAE and the clustering used in self-supervised learning. Based on this reformulation, we propose to use an online clustering method to address the codebook collapse issue of training VQ-VAE, by adding the equipartition constraint from Asano et al. (2020) and Caron et al. (2020). The online clustering method, inspired by the OT techniques used in Caron et al. (2020), assigns to each cluster (represented by a code in the codebook) the same number of samples (see Figure 1). Then, we use a Gumbel-softmax trick Jang et al. (2017) to sample from the discrete categorical distribution while easily assessing its gradient. The resulting approach, named OT-VAE, enforces the full utilization of the codebook while not using any heuristics, namely stop-gradient, EMA, and codebook reset. Unlike SQ-VAE that uses a stochastic quantization and dequantization process, our approach explicitly enforces the equipartition constraint for quantization in a deterministic way while only using a stochastic decoding. To the best of our knowledge, our approach shows the first time that such an equipartition condition, arised in the field of self-supervised learning, is also useful for generative tasks. We empirically validate our approach on three different data modalities: images, speech and 3D human motions. For all the modalities, OT-VAE shows better reconstruction with higher perplexity than other VQ-VAE variants on several datasets. In particular, OT-VAE achieves state-of-the-art results on the AIST++ (Li et al., 2021) dataset for 3D dance generation. Overall, our contribution can be summarized below:

- We reformulate VQ-VAE as an instance of WAEs, which provides a connection between distribution matching in the latent space of VQ-VAE and the clustering methods used in self-supervised learning.

- We propose OT-VAE, a novel unsupervised generative model explicitly using the equipartition constraint with OT to address the codebook collapse issue in VQ-VAE.

- In our experiments, without using classic heuristics (such as stop-gradient, EMA, codebook reset etc.), we show that OT-VAE achieves better reconstruction and perplexity than other variants of VQ-VAE for three data modalities: image, speech and 3D human motion.

- Using OT-VAE instead of VQ-VAE in the Bailando model (Li et al., 2022), we obtain state-of-the-art results for 3D Dance generation. Precisely, we improve $FID_k$ from 28.75 to 26.74 and $FID_g$ from 11.82 to 9.81 on the AIST++ dataset.

## 2 RELATED WORK

**VQ-VAE** VQ-VAE framework was first introduced in Van Den Oord et al. (2017), as a variant of VAE (Kingma & Welling, 2014) with a discrete prior. VQ-VAE shows good performance on various generation tasks, which includes: image synthesis (Williams et al., 2020; Esser et al., 2021), text to image generation (Ramesh et al., 2021), motion generation (Li et al., 2022), music generation (Dieleman et al., 2018; Dhariwal et al., 2020) etc. However, a naive training of VQ-VAE suffers from the codebook collapse. To alleviate the problem, a number of techniques are commonly used during the training, including stop-gradient along with some losses (Van Den Oord et al., 2017),

EMA for codebook update (Van Den Oord et al., 2017), codebook reset (Williams et al., 2020), etc. Our work is highly related to SQ-VAE (Takida et al., 2022), which also aimed at improving codebook utilization without using those heuristics. SQ-VAE proposed to perform stochastic quantization and dequantization at the early stage of the training and gradually anneal the process toward a deterministic one. However, a proper prior distribution needs to be carefully chosen for different tasks. In contrast, our work uses the same prior throughout different tasks and data modalities.

**Wasserstein autoencoders**  WAEs (Tolstikhin et al., 2018) consist of a class of generative models based on the reconstruction from an autoencoder and a regularizer in the latent space that encourages the training distribution to match the prior. The regularizer is defined through a divergence between probability distributions. In the previous research, various divergence functions have been proposed, such as MMD (Tolstikhin et al., 2018), GAN (Makhzani et al., 2016), Sliced Wasserstein (Kolouri et al., 2018), Sinkhorn (Patrini et al., 2020), and so on. While all of the WAE variants use a continuous prior, our work shows that VQ-VAE could be interpreted as an instance of WAEs with a discrete prior. By using the Sinkhorn divergence proposed in Patrini et al. (2020), our work suggests a natural connection between VQ-VAE and clustering-based self-supervised learning methods.

**Clustering-based self-supervised learning**  The matching problem between the posterior distribution and the discrete prior distribution is related the clustering-based methods for self-supervised learning (Li et al., 2020b; Alwassel et al., 2020). The seminal work by Caron et al. (2018) shows that the assignments obtained by K-means can be used as pseudo-labels to learn discriminative representations for images. However, their method could suffer from the cluster collapse issue caused by K-means, where some clusters are collapse to a single entity during training. To address this issue, Asano et al. (2020) propose to incorporate an equipartition constraint and cast it as an OT problem. Then, Caron et al. (2020) propose a Sinkhorn-Knopp (Cuturi, 2013) based online algorithm, making it scalable to very large datasets. Our work shows the connection between VQ-VAE and these clustering-based self-supervised learning methods and demonstrate the first time that these clustering techniques are also useful for generative tasks.

## 3  BACKGROUND ON VQ-VAE

VQ-VAE (Van Den Oord et al., 2017) consists of deterministic autoencoders that allow learning a discrete representation of the data. The main building blocks of VQ-VAE are similar to any autoencoders with an additional codebook. The codebook $C$ is defined as a set of $K$ trainable vectors $c_1, \ldots, c_K$ in $\mathbb{R}^{d_h}$. We denote by $C^{d_z} \subset \mathbb{R}^{d_z \times d_h}$ a $d_z$-tuple related to the codebook $C$ with $d_z$ describing the number of components in the latent space. For a latent variable $Z \in C^{d_z}$, we denote by $z_i \in C$ its $i$-th entry. The deterministic encoder first maps an observation $X \in \mathcal{X}$ to the same space as $Z$ using a neural network $h_\phi : \mathcal{X} \to \mathbb{R}^{d_z \times d_h}$ with trainable parameters $\phi$, followed by a quantization step projecting $h_\phi(X)$ onto $C^{d_z}$. The quantization process is modelled as a deterministic categorical posterior distribution such that $\hat{Z}_i(X) := \arg\min_{c_k \in C} \|h_\phi(X)_i - c_k\|_2^2$. Then the objective of VQ-VAE is defined as

$$\mathcal{L}_{\text{VQ-VAE}} := -\log p_\theta(X|\hat{Z}(X)) + \|\text{sg}(h_\phi(X)) - \hat{Z}(X)\|^2 + \beta \|h_\phi(X) - \text{sg}(\hat{Z}(X))\|^2, \quad (1)$$

where $p_\theta$ is a decoder with parameters $\theta$, sg denotes the stop gradient operator, and $\beta$ is a hyperparameter. The hard assignment in the quantization process makes the training of VQ-VAE sensitive to the initialization and prone to local minima. To improve the utilization of the codebook, various strategies have been adopted in previous work, often involving a huge amount of work for hyperparameter tuning.

## 4  METHOD

In this section, we reformulate VQ-VAE as an instance of WAEs. The new formulation offers a natural choice of the prior distribution and the objective function. This yields a simple yet effective autoencoder framework that can learn discrete representations of the data.

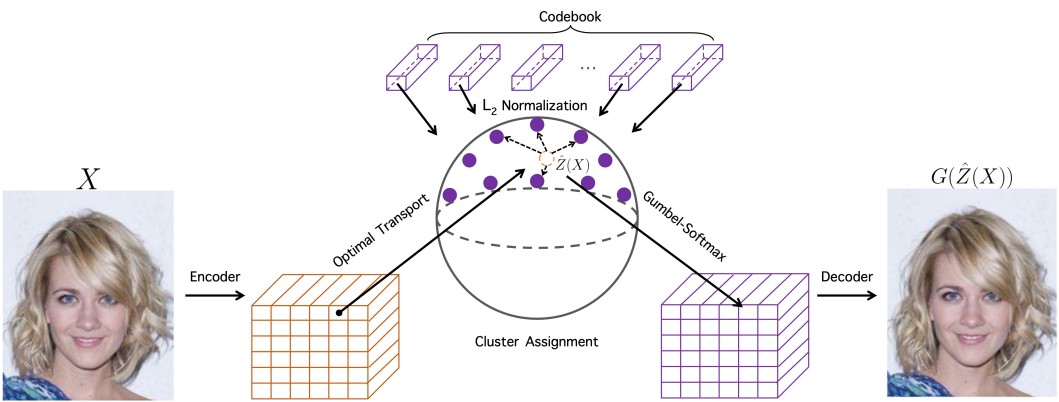

Figure 2: Overview of OT-VAE: the encoder maps a sample $X$ to the soft cluster assignment variables $\hat{Z}(X)$ given by an OT based clustering algorithm. Then, $\hat{Z}(X)$ is fed to a stochastic decoder, consisting of a Gumbel-softmax sampler and a deterministic decoder, to reconstruct the data sample.

### 4.1 VQ-VAE AS AN INSTANCE OF WASSERSTEIN AUTOENCODERS

Consider an observation $X$ with the true data distribution $P_X$ on the input space $\mathcal{X}$. Let us denote by $P_Z$ a fixed prior distribution on a latent space $\mathcal{Z}$. A sample from the prior distribution $Z$ is mapped to the input space $\mathcal{X}$ with a transformation parametrized by a neural network $G : \mathcal{Z} \to \mathcal{X}$. If we denote by $P_G$ the induced marginal distribution, then the OT cost between $P_X$ and $P_G$ has been shown by Tolstikhin et al. (2018) to take a simpler form:

$$W_c(P_X, P_G) := \inf_{\Gamma \in \Pi(P_X, P_G)} \mathbb{E}_{(X,Y)\sim\Gamma}[c(X,Y)] = \inf_{Q:Q_z=P_Z} \mathbb{E}_{X\sim P_X} \mathbb{E}_{Z\sim Q(Z|X)}[c(X, G(Z))], \tag{2}$$

where $c$ is a cost function such as the $L_2$ cost, $\Pi(P_X, P_G)$ denotes the admissible couplings between $P_X$ and $P_Z$, and we consider a conditional distribution (the encoder) $Q(Z|X)$ such that its $Z$ marginal $Q_Z := \mathbb{E}_{X\sim P_X}[Q(Z|X)]$ is identical to the prior distribution $P_Z$. Tolstikhin et al. (2018) showed that learning the generative model $G$ amounts to solving the following objective:

$$\min_G \min_{Q(Z|X)} \mathbb{E}_{X\sim P_X} \mathbb{E}_{\hat{Z}\sim Q(Z|X)}[c(X, G(\hat{Z}))] + \lambda \mathcal{D}(Q_Z, P_Z), \tag{3}$$

where $\lambda > 0$ is a Lagrangian multiplier and $\mathcal{D}$ is an arbitrary divergence between probability distributions on $Z$ that controls the discrepancy between the prior and the learned conditional distribution. The resulting objective and models are called Wasserstein autoencoders (WAE).

VQ-VAE could also be interpreted as an instance of WAEs with a discrete prior distribution. Let us denote by $\mathrm{Cat}(K, 1/K)$ the uniform categorical distribution such that $\mathbb{P}(x = j) = 1/K$ for $x \sim \mathrm{Cat}(K, 1/K)$ and $j = 1, \dots, K$. As noticed by Van Den Oord et al. (2017), the prior distribution $P_Z$ is assumed to be defined on the latent discrete space $\{1, \dots, K\}^{d_z}$ with $d_z$ i.i.d. random variables following the uniform categorical distribution such that $\mathbb{P}(Z_i = j) = 1/K$ for any $Z \sim P_Z$. Now let us consider a deterministic encoder $h_\phi$ that encodes each input $X \in \mathcal{X}$ to $h_\phi(X) \in \mathbb{R}^{d_z \times d_h}$. We define the posterior distribution as $\hat{Z}_i(X) = \arg\min_{k\in\{1,\dots,K\}} \|h_\phi(X)_i - c_k\|_2^2$ the nearest code index. When $Q(Z|X)$ takes the form of $\hat{Z}(X)$ and $\mathcal{D}$ is the KL divergence, the second term in Equation 3 is constant and we recover the objective of VQ-VAE after adding the hard assignment constraint in the posterior. Despite its simplicity and deterministic formulation, the main limitation of VQ-VAE is that it does not explicitly enforce the equal partition constraint introduced in the prior. In order to fully match encoded training distribution to the prior, we propose to relax the hard assignment and use an optimal transport based divergence.

### 4.2 OPTIMAL TRANSPORT QUANTIZED AUTOENCODERS

Without loss of generality, we assume $d_z = 1$ in the following and one could generalize our analysis for $d_z > 1$ by assuming $h_\phi(X)_i$ to be independent, which was also used in Van Den Oord et al.

(2017). Instead of using the hard assignment for the posterior distribution, we propose to use a soft assignment

$$Q(Z|X) = \hat{Z}(X) := \text{softmax}_{k=1,\ldots,K}(-\|h_\phi(X) - c_k\|_2^2) \in \Delta^K, \tag{4}$$

where $\Delta^K$ denotes the probability simplex of dimension $K-1$. If we sample $x_1, \ldots, x_n$ i.i.d. from $P_X$, $Q_Z$ could be considered as a discrete measure evenly distributed on $\hat{z}_1, \ldots, \hat{z}_n$ where $\hat{z}_j := \hat{Z}(x_j) \in \Delta^K$. On the other hand, the prior distribution is also a discrete measure such that $Z = \sum_k \frac{1}{K}\delta_{u_k}$ where $u_k \in \Delta^K$ denotes the binary vector with only its $k$-th entry equal to 1 using the one-hot representations. Then, the discrete optimal transport cost between the corresponding $Q_Z$ and $P_Z$ is given by

$$\text{OT}(Q_Z, P_Z) := \min_{\Gamma \in \Pi(1/n, 1/K)} \sum_{j=1}^n \sum_{k=1}^K \Gamma_{jk} c(\hat{z}_j, u_k) = \min_{\Gamma \in \Pi(1/n, 1/K)} \sum_{j,k} -\Gamma_{jk} \log \hat{z}_{jk}, \tag{5}$$

where $c$ is the KL divergence on $\Delta^K$ such that $c(p, q) = \sum_i p_i \log(p_i/q_i)$ and $\Pi$ is the transportation polytope of the admissible couplings between $Q_Z$ and $P_Z$, commonly known in the literature on optimal transport (Peyré et al., 2019)

$$\Pi(1/n, 1/K) := \left\{ \Gamma \in \mathbb{R}_+^{n \times K} : \Gamma \mathbf{1} = \frac{1}{n}, \Gamma^\top \mathbf{1} = \frac{1}{K} \right\}. \tag{6}$$

Thus, we recover the self-labeling scheme with the equal partition constraint developed in the field of self-supervised learning (Asano et al., 2020). In particular when we project both $h_\phi(X)$ and prototypes $c_k$ onto the unit $L^2$-sphere, we recover the formulation of Caron et al. (2020), which amounts to solving an online OT problem for samples within a minibatch as detailed in Section 4.2.1. The only difference is that we perform clustering on the set of small components of the data, such as patches for images, while Caron et al. (2020) clustered the features of the entire images. After the clustering process, a latent variable can be sampled from the categorical distribution given by $p_j$ using the Gumbel-softmax relaxation, which will be detailed in Section 4.2.2.

Another way to derive the OT regularization in Equation 5 is to consider a parametrized prior distribution instead of the fixed categorical distribution. Specifically, we consider a mixture of $K$ Dirac measures with equal mixing masses $Z_C := \sum_{k=1}^K \frac{1}{K}\delta_{c_k}$. Now, we consider $\hat{Z}_C(X) := h_\phi(X)$ as the posterior distribution $Q(Z|X)$. Then, the Wasserstein distance between $Q_Z$ and $P_Z$ is equal to

$$W_2^2(Q_Z, P_Z) := \min_{\Gamma \in \Pi(1/n, 1/K)} \sum_{j,k} \Gamma_{jk} \|h_\phi(x_j) - c_k\|^2, \tag{7}$$

which admits the same solution as Equation 5 since the denominator of $\hat{z}_{jk}$ does not depend on $k$. This formulation suggests that $C$ could be regarded as the Wasserstein Barycenter of $Q(Z|X)$ (Cuturi & Doucet, 2014). The resulting autoencoder could thus be simply interpreted as a Sinkhorn Autoencoder (Patrini et al., 2020) when using the entropic regularization to the OT problem.

Based on the above observations, the final objective of the Optimal Transport Quantized Autoencoders (OT-VAE) can be written as

$$\mathcal{L}_{\text{OT-VAE}} := \min_G \min_{Q(Z|X)} \underbrace{\mathbb{E}_{X \sim P_X} \mathbb{E}_{\hat{Z} \sim Q(Z|X)}[c(X, G(\hat{Z}))]}_{\mathcal{L}_{re}} + \lambda \underbrace{\text{OT}(Q_Z, P_Z)}_{\mathcal{L}_{ot}}, \tag{8}$$

where the first term corresponds to the reconstruction loss $\mathcal{L}_{re}$ and the second term $\mathcal{L}_{ot}$ amounts to clustering the components of the data in the latent space under the equipartition constraint. An overview of OT-VAE is illustrated in Figure 2.

### 4.2.1 ONLINE CLUSTERING WITH EQUIPARTITION CONSTRAINT

Here, we provide details about how to compute the OT loss between $Q_Z$ and $P_Z$ in an online fashion following the techniques in Caron et al. (2020). In order to make the clustering algorithm online, we compute the the cluster assignments $\Gamma$ using the features of data components (*e.g.* image patch features) within a batch. We thus assume $n$ to be the number of component features within a batch and reuse the previous notations. We also assume $\hat{h}_j := h_\phi(x_j)$ and $c_k$ to have unit $L_2$-norms

as in Yu et al. (2021) for generative modeling and Caron et al. (2020) for self-supervised learning. Then, by adding an entropic regularization, the OT loss in Equation 7 is equivalent to solving Caron et al. (2020)

$$\min_{\Gamma \in \Pi(1/n, 1/K)} \sum_{j,k} \Gamma_{jk} D_{jk} - \varepsilon H(\Gamma), \tag{9}$$

where $D_{jk} = -\hat{h}_j^\top c_k$ and $H(\Gamma) = -\sum_{jk} \Gamma_{jk}(\log \Gamma_{jk} - 1)$ is the entropic regularization with a parameter $\varepsilon > 0$. This problem can be efficiently solved by an iterative matrix scaling algorithm known as Sinkhorn-Knopp algorithm. More details can be found in Section A.1 of the Appendix.

Once a solution $\Gamma^\star$ to the above problem is found, we can inject it into the OT loss in Equation 5 and obtain:

$$\text{OT}(Q_Z, P_Z) = -\sum_{j=1}^{n} \sum_{k=1}^{K} \Gamma_{jk}^\star \log \hat{z}_{jk,\tau}, \text{ where } \hat{z}_{jk,\tau} := \frac{\exp(1/\tau \hat{h}_j^\top c_k)}{\sum_{k'=1}^{K} \exp(1/\tau \hat{h}_j^\top c_{k'})}, \tag{10}$$

where we added a temperature parameter $\tau$ to $\hat{z}_j$, similar to Caron et al. (2020). This loss function is jointly minimized with the reconstruction loss in Equation 8 with respect to the autoencoder parameters and the codebook $C$. In our experiments, $\tau$ is a learnable parameter with its initial value as a hyperparameter.

### 4.2.2 GUMBEL-SOFTMAX RELAXATION

Once the input $x_j$ is mapped to the soft assignment $\hat{z}_j \in \Delta^K$ in Equation 4, we need to find a way to sample from this categorical distribution with a gradient estimator of the parameters. A simple approximation is the Gumbel-softmax relaxation (Jang et al., 2017; Maddison et al., 2017), which has also been used in the training of VQ-VAE (Esser et al., 2021) or its variants (Takida et al., 2022). Specifically, the Gumbel-softmax function is defined as a vector $q \in \Delta^K$ such that

$$q_k(\hat{z}_j) := \frac{\exp(1/\tau'(\log \hat{z}_{jk} + g_k))}{\sum_{k'=1}^{K} \exp(1/\tau'(\log \hat{z}_{jk'} + g_{k'}))} = \frac{\exp(1/\tau'(\hat{h}_j^\top c_k + g_k))}{\sum_{k'=1}^{K} \exp(1/\tau'(\hat{h}_j^\top c_{k'} + g_{k'}))} \text{ for } k = 1, \ldots, K, \tag{11}$$

providing a continuous differentiable approximation to a sample drawn from the categorical distribution with class probabilities $\hat{z}_{jk}$ when the temperature parameter $\tau'$ approaching 0. Here, $g_1, \ldots, g_K$ are i.i.d. samples drawn from $\text{Gumbel}(0, 1)$. In practice, we adopt the annealing strategy for $\tau'$ as in Jang et al. (2017), by starting with a high temperature and annealing to a small but non-zero value temperature. In such a way, the gradients at $\hat{h}_j$ can be easily back-propagated to the encoder using any deep learning framework allowing automatic differentiation.

### 4.3 PRIOR LEARNING WITH AUTOREGRESSIVE MODELS

Once the discrete representations are learned with OT-VAE, one could use a deep autoregressive model for generation tasks, which has shown state-of-the-art performances on different tasks (Dieleman et al., 2018; Esser et al., 2021; Ramesh et al., 2021; Dhariwal et al., 2020). The principal idea is to learn the probability of the encoded discrete representations through the chain rule of probability: $p(Z) = p(Z_1) \prod_{i=2}^{n} p(Z_i | Z_{1:i-1})$. The conditional probability can be parameterized by a Generative Pre-trained Transformer (GPT) (Vaswani et al., 2017; Radford et al., 2018).

## 5 EXPERIMENTS

In this section, we empirically validate OT-VAE on 3 different data modalities: images, speech and 3D human motions. For images, we use CelebA (Liu et al., 2015) and CelebAHQ-Mask (Liu et al., 2015), which represent continuous and discrete data distributions respectively. For speech, we use ZeroSpeech 2019 (Dunbar et al., 2019). Finally, we use AIST++ (Li et al., 2021) for 3D dance motion generation.

### 5.1 IMAGES

**Datasets CelebA** is a face dataset proposed in Liu et al. (2015). Following SQ-VAE (Takida et al., 2022), we apply OT-VAE on the aligned and cropped version of the dataset, which consists of

Table 1: **Evaluation on the test set of CelebA (Liu et al., 2015)**. We report MSE ($\times 10^{-3}$). Following SQ-VAE (Takida et al., 2022), the codebook contains 512 codes and each code is with dimension 64. Experiments of reconstruction are repeated *three* times.

| Method | MSE ($\times 10^{-3}$) $\downarrow$ | Perplexity $\uparrow$ | Generation FID $\downarrow$ |
|---|---|---|---|
| VAE | $4.79 \pm 0.01$ | - | - |
| VQ-VAE + EMA | $1.33 \pm 0.41$ | - | - |
| VQ-VAE + EMA + Code Reset | $1.62 \pm 0.36$ | - | - |
| Gaussian SQ-VAE | $0.96 \pm 0.00$ | $413.2 \pm 4.9$ | 20.8 |
| OT-VAE (Ours) | $\mathbf{0.94 \pm 0.00}$ | $\mathbf{433.3 \pm 5.1}$ | 20.0 |

Table 2: **Evaluation on the test set of CelebAHQ-Mask (Liu et al., 2015)**. We report pixel error (%), mIoU, and perplexity. Following SQ-VAE (Takida et al., 2022), the codebook contains 64 codes and each code is with dimension 64. Experiments are repeated *three* times.

| Method | Pixel error (%) $\downarrow$ | mIoU $\uparrow$ | Perplexity $\uparrow$ |
|---|---|---|---|
| VAE | $8.79 \pm 0.01$ | $55.8 \pm 0.3$ | - |
| VQ-VAE + EMA | $6.95 \pm 0.14$ | $59.7 \pm 0.7$ | $46.2 \pm 2.0$ |
| NC SQ-VAE | $6.63 \pm 1.38$ | $64.1 \pm 5.4$ | $12.6 \pm 5.2$ |
| vMF SQ-VAE | $3.51 \pm 0.17$ | $74.6 \pm 0.0$ | $52.4 \pm 0.8$ |
| OT-VAE (Ours) | $\mathbf{3.40 \pm 0.08}$ | $\mathbf{75.5 \pm 0.8}$ | $\mathbf{61.1 \pm 1.0}$ |

202,599 images split into 162,770, 19,867 and 19,962 for train, validation and test. **CelebAHQ-Mask** contains annotations of face attributes of CelebA, which is thus a categorical image dataset with 19 categories. There are 24,183, 2,993 and 2,824 images respectively in the train, validation and test set. For both datasets, we adopt the same pre-processing as SQ-VAE and report performances on the test set with the model performing the best on the validation dataset.

**Evaluation metric** For continuous data representation, *i.e.* CelebA, we report Mean Squared Error (MSE) to quantify the reconstruction quality. For discrete data representation as CelebAHQ-Mask, the reconstruction quality is measured using the percentage of pixels that are incorrectly predicted (Pixel error), and the mean of the class-wise intersection over union (mIoU). We also report the perplexity on the test set, which can be considered as a measure of the codebook utilization. The perplexity of a distribution $p$ is given by the exponential of its entropy: $\mathrm{PPL}(p) = \prod_x p(x)^{-p(x)}$.

**Implementation details** We follow the evaluation protocol provided by SQ-VAE (Takida et al., 2022) [1] and keep the same backbone architecture of OT-VAE as SQ-VAE. For both datasets, the hyper-paramters of OT-VAE are set as $\lambda$=1e-3 (regularization weight in Equation 8) and $\log {}^1\!/_\tau = 1$ (initial temperature in Equation 10). The ablation study of these hyper-parameters on CelebA (Liu et al., 2015) are provided in Appendix C. We also provide more implementation details in Appendix D, which includes training schema, architectures, hyper-parameters etc.

**Results** The results are provided in Table 1 and 2 for CelebA (Liu et al., 2015) and CelebAHQ-Mask (Liu et al., 2015) respectively. We repeat each experiment of reconstruction three times and report the average and standard deviation for each metric. For generation on CelebA, we use a standard GPT as the generative model. More details can be found in the Appendix D. For both continuous and discrete data distributions, our OT-VAE achieves clearly better reconstruction quality than other variants of VQ-VAE. In particular, the proposed OT-VAE obtains higher perplexity which suggests a more efficient usage of the codebook capacity than SQ-VAE (Takida et al., 2022). Note that SQ-VAE requires selecting proper priors for different tasks. Using either Naive categorical (NC SQ-VAE) or von Mises–Fisher (vMF SQ-VAE) as the prior, the performance vary significantly on CelebAHQ-Mask. In contrast, the proposed OT-VAE does not need any selection of priors and its performances are obtained with the same hyper-parameters, demonstrating its superior robustness. Our approach also results in better performance for generation, compared to SQ-VAE. Note that computing the coupling matrix takes extra computation for training. Precisely, training OT-VAE takes about 102s for one epoch on CelebA, while training a standard VQ-VAE only takes about 89s. For inference, both methods take the same time as the OT coupling matrix is not needed.

---

[1]SQ-VAE: https://github.com/sony/sqvae/tree/main/

Table 3: **Evaluation on the test set of ZeroSpeech2019 (Dunbar et al., 2019)**. We report MSE (dB$^2$), perplexity, FDSD and KDSD. Following SQ-VAE (Takida et al., 2022), the codebook contains 512 codes of dimension 64. Experiments of reconstructions are repeated *three* times. 'r' and 'g' indicate resynthesis and generation respectively.

| Method | | | ZeroSpeech2019 | | | |
|---|---|---|---|---|---|---|
| | MSE (dB$^2$)↓ | Perplexity↑ | rFDSD↓ | rKDSD↓ | gFDSD↓ | gKDSD↓ |
| Natural speech | - | - | 5.82 | 3.03 | 5.82 | 3.03 |
| VQ-VAE + EMA | $34.33 \pm 1.57$ | - | - | - | - | - |
| Gaussian SQ-VAE | $32.13 \pm 1.78$ | $440.0 \pm 2.7$ | $8.95 \pm 1.11$ | $14.55 \pm 1.98$ | 10.90 | 15.09 |
| OT-VAE (Ours) | $\textbf{28.73} \pm \textbf{0.29}$ | $\textbf{446.2} \pm \textbf{6.5}$ | $\textbf{7.29} \pm \textbf{0.44}$ | $\textbf{11.61} \pm \textbf{0.78}$ | **10.60** | **14.55** |

## 5.2 SPEECH

**Datasets** ZeroSpeech2019 (Dunbar et al., 2019) is a multi-speaker corpus sampled at 16000 Hz. Following SQ-VAE (Takida et al., 2022), we use the subset *train_voice* and *train_unit* as the train set (~20.3 hours), and *train_parallel_voice* for the test set (~10 mins). We randomly sample 200 samples from the train set as a validation set to select the best model. The number of total speakers for both train and test is 102. We apply the same pre-processing as in SQ-VAE (Takida et al., 2022) to extract 80-dimensional log-mel spectrogram features for all the speech data. We provide more details in Appendix E.

**Evaluation metric** Following SQ-VAE (Takida et al., 2022), we report MSE in dB$^2$ to measure the reconstruction quality and the perplexity for codebook utilization.

**Implementation details** We follow the evaluation protocol provided by SQ-VAE (Takida et al., 2022) and keep the backbone architecture of OT-VAE same as SQ-VAE. The hyper-paramters of OT-VAE are set as $\lambda$=1e-3 (regularization weight in Equation 8) and $\log 1/\tau = 5$ (initial temperature in Equation 10). The ablation study of these hyper-parameters on ZeroSpeech2019 (Dunbar et al., 2019) is provided in Appendix C. More implementation details such as training schema, architectures, and hyper-parameters are given in Appendix E.

**Results** We show the results in Table 3. Same as SQ-VAE (Takida et al., 2022), we repeat each experiment on reconstruction three times and report the mean and standard deviation for each metric. For generation, we use a standard GPT, more details can be found in the Appendix E. Our proposed OT-VAE significantly outperforms SQ-VAE (Takida et al., 2022) both in terms of the reconstruction quality (lower value both on MSE, FDSD and KDSD), the codebook utilisation (6.2 higher in perplexity), and generation quality (lower FDSD and lower KDSD).

## 5.3 3D HUMAN MOTIONS

**Dataset** In order to fully explore the capacity of OT-VAE, we also conduct experiments for a challenging task, namely 3D dance generation. We use AIST++ (Li et al., 2021), which is the largest publicly available dataset for this task. The dataset contains 992 pieces paired music-motion sequences, where 952 are kept for training and 40 are used for evaluation. Motion sequences are provided in the SMPL (Loper et al., 2015) format with 60 FPS. Following Li et al. (2022), we use the 3D coordinates of 24 joints to represent motions. The task aims to generate motions conditioned on music sequences. We compare our method to the state-of-the-art method (Li et al., 2022), based on VQ-VAE. We provide more details about the baseline Li et al. (2022) in Appendix F.

**Evaluation metric** We measure the dance quality with Frechet Inception Distances (FID) (Heusel et al., 2017) between the predicted motion and all motion sequences (including training and test data) of the AIST++ dataset. Following Li et al. (2021; 2022), we consider two features: kinetic features (Onuma et al., 2008) (denoted as 'k') and geometric features (Müller et al., 2005) (denoted as 'g'). We extract both features using fairmotion [2]. Following Li et al. (2021; 2022), we also report diversity, which is the average feature euclidean distance of generated motions. We additionally include the perplexity to measure the codebook utilization.

**Implementation details** We keep the same CNN and GPT architectures as in Li et al. (2022) for discrete representation learning and generation respectively. We train the proposed OT-VAE using

---

[2]fairmotion: https://github.com/facebookresearch/fairmotion

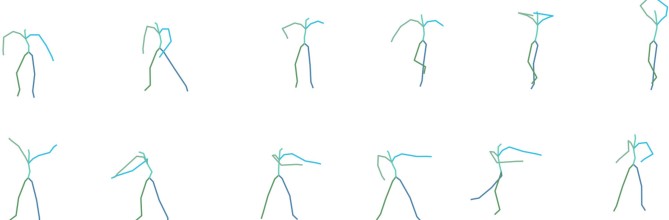

Figure 3: Two samples of dance motion are generated with the proposed OT-VAE, which are conditioned on different pieces of music from the test set of AIST++ (Li et al., 2022).

Table 4: **Evaluation on the test set of AIST++ (Li et al., 2021) for 3D dance generation**. $FID_k$ and $FID_g$ are used to evaluate generation quality. We also report $DIV_k$ and $DIV_g$ to quantify motion diversity for completeness. Following Li et al. (2022), we train two codebooks for upper and lower body respectively. Each codebook contains 512 codes with dimension 512.

| Method | Motion Quality | | Motion Diversity | | Perplexity ↑ |
|---|---|---|---|---|---|
| | $FID_k \downarrow$ | $FID_g \downarrow$ | $DIV_k \uparrow$ | $DIV_g \uparrow$ | |
| Ground Truth | 17.10 | 10.60 | 8.19 | 7.45 | - |
| **Motion Reconstruction** | | | | | |
| VQ-VAE + EMA + Code Reset (Li et al., 2022) | 28.23 | 12.63 | 6.80 | 6.57 | 138.14 |
| OT-VAE (Ours) | **23.50** | **9.42** | **6.98** | 6.47 | **243.21** |
| **Motion Generation Approaches** | | | | | |
| Li *et al.* (Li et al., 2020a) | 86.43 | 43.46 | 6.85 | 3.32 | - |
| DanceNet (Zhuang et al., 2022) | 69.18 | 25.49 | 2.86 | 2.85 | - |
| DanceRevolution (Huang et al., 2021) | 73.42 | 25.92 | 3.52 | 4.87 | - |
| FACT (Li et al., 2021) | 35.35 | 22.11 | 5.94 | 6.18 | - |
| **Motion Generation with GPT** | | | | | |
| VQ-VAE + EMA + Code Reset (Li et al., 2022) | 28.75 | 11.82 | 6.41 | **6.13** | 166.84 |
| OT-VAE (Ours) | **26.74** | **9.81** | **6.52** | 5.89 | **334.53** |

the same reconstruction loss as in Li et al. (2022), measuring the reconstruction between the input and reconstructed motion as well as their velocity and acceleration. The hyper-parameters are set as $\lambda$=1e-3 (regularization weight in Equation 8) and $\log 1/\tau = 1$ (initial temperature in Equation 10). Note that the VQ-VAE used in Li et al. (2022) is trained using the stop-gradient operator, EMA and codebook reset. We provide more implementation details in Appendix F, which includes training schema, architectures, hyper-parameters.

**Results** Metrics for both reconstruction and generation are shown in Table 4. For reconstruction, the proposed OT-VAE achieves better performance than Li et al. (2022) which uses a standard VQ-VAE. Note that numerous heuristics are used in Li et al. (2022), while non of these heuristics are employed in OT-VAE and we achieve better reconstruction quality and higher perplexity.

The downstream generation task also benefits from the better reconstruction. Using the same GPT architecture, our OT-VAE achieves state-of-the-art on this task and outperforms VQ-VAE in Li et al. (2022) in terms of motion quality and achieves comparable diversity. We showcase and illustrate two samples of generated motion in Figure 3, which demonstrates the high-quality motion generated by our algorithm. In the supplementary material, we additionally provide videos containing dance motion along with the input music.

## 6 CONCLUSION

We proposed OT-VAE, a WAE-based generative model with a discrete prior and a Sinkhorn divergence to match the encoded training distribution and the prior. Our approach first shows the clustering techniques developed in the field of self-supervised learning could be beneficial for generative tasks. With the rapid development of self-supervised learning in recent years, we believe that more algorithms arising from self-supervised learning, such as contrastive strategies (Jaiswal et al., 2020), knowledge distillation (Grill et al., 2020; Caron et al., 2021) or a direct use of large-scale pre-trained models could be adaptable for generative modeling.

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

# Appendix

In the appendix, we provide the following content:

1. Additional background in Appendix A, including Sinkhorn-Knopp Algorithm.

2. Pytorch implementation of OT-VAE in Appendix B.

3. Analysis on $\lambda$ and $\tau$ (regularization weight and initial temperature of $\mathcal{L}_{OT}$) for OT-VAE on the test set of CelebA (Liu et al., 2015) and ZeroSpeech2019 (Dunbar et al., 2019) in Appendix C.

4. Details of experiments on images in Appendix D.

5. Details of experiments on speech data in Appendix E.

6. Details of experiments on 3D dance generation in Appendix F.

Additionally, in the supplementary material (the uploaded zip file, `suppMat_OTVAE.zip`), we also provide the source code to reproduce experiments on images in the folder `./suppMat_OTVAE/code.zip` and videos of generated dance in `./suppMat_OTVAE/visual_videos.zip`.

## A    ADDITIONAL BACKGROUND

This section provides additional background including the details about the Sinkhorn-Knopp algorithm.

### A.1    SINKHORN-KNOPP ALGORITHM

The online clustering problem with the equipartition constraint is equivalent to the entropy-regularized OT problem in Equation 9, given by

$$\min_{\Gamma \in \Pi(1/n, 1/K)} \langle \Gamma, D \rangle_F - \varepsilon H(\Gamma), \tag{12}$$

where $H(\Gamma) := -\sum_{ij} \Gamma_{ij}(\log \Gamma_{ij} - 1)$ and $D_{jk} = -\hat{h}_j^\top c_k$ is the cost matrix. This problem can be efficiently solved by the Sinkhorn-Knopp algorithm, which is an iterative matrix scaling method to approach the double stochastic matrix. Specifically, the $\ell$-th iteration of the algorithm performs the following updates:

$$u^{(\ell)} = \frac{1/n}{Sv^{(\ell)}}, \qquad v^{(\ell+1)} = \frac{1/K}{S^\top u^{(\ell)}},$$

where $S = e^{-D/\varepsilon}$. Then, the matrix $\mathrm{diag}(u^{(\ell)}) S \mathrm{diag}(v^{(\ell)})$ converges to the solution of Equation 12 when $\ell$ tends to $\infty$. This algorithm converges faster with larger $\varepsilon$ as strong regularization leads to a more convex objective (Peyré et al., 2019). However when $\varepsilon$ becomes too small, some of the elements of the denominators become null and result in a division by 0. To overcome this stability issue, it is preferable to do the computations in the log-scale (Peyré et al., 2019). This algorithm can be easily adapted to a batch of encoded data features $\hat{h}$, leading to a scalable and GPU-friendly computation. In our experiments, we also observed that only few iterations (5 or 10) are sufficient to obtain good performance. A Pytorch implementation of this algorithm is provided in Section B.

## B    PYTORCH IMPLEMENTATION OF OT-VAE

```
1  # C : codebook
2  # Y : features from the encoder}
3  # Z : features after ot quantization}
4  # tau: learnable temperature parameter in OT
```

```
5
6  import torch
7  import torch.nn.functional as F
8  import math
9
10 # ---> Sinkhorn-Knopp algorithm <---
11 def Log_Sinkhorn(K, eps = 0.5):
12     m, n = K.shape
13     v = K.new_zeros((m,))
14     a, b, K = 0, math.log(m / n), K / eps
15     for _ in range(10):}
16         u = -torch.logsumexp(v.view(m, 1) + K, dim=0) + b
17         v = -torch.logsumexp(u.view(1, n) + K, dim=1) + a
18     return torch.exp(K + u.view(1, n) + v.view(m, 1))
19
20
21 # ---> Gumbel-Softmax Sample algorithm <---
22 def Gumbel_Softmax_Sample(logit, gumbel_temperature) :
23     #Sampling U from a uniform distribution on [0, 1)
24     U = torch.rand(logit)
25     sample = logit - torch.log(-torch.log(U))
26     return F.softmax(sample / gumbel_temperature, dim=-1)
27
28 # ---> L2 Normalization <---
29 Y_norm = F.normalize(Y, p=2.0)
30 C_norm = F.normalize(C, p=2.0)
31
32 #---> Compute probability <---
33 logit = Y_norm.mm(C_norm.T) * tau.exp()
34 probs = torch.softmax(logit, dim=-1)
35 log_probs = torch.log_softmax(logit, dim=-1)
36
37 # ---> Optimal Transport <---
38 with torch.no_grad() :
39     q_ot = Log_Sinkhorn(logit)
40
41 # ---> Training <---
42 if is_training:
43     sampling = Gumbel_Softmax_Sample(logit, gumbel_temperature)
44 # ---> Evaluation <---
45 else:
46     sampling = torch.argmax(probs, dim=-1)
47
48 Z = torch.mm(sampling, C)
49
50 # ---> OT Loss <---
51 loss_ot = -torch.mean(q_ot, log_probs)
```

## C    ABLATION STUDY

In this section, we provide analysis on $\lambda$ and $\tau$ (regularization weight and initial temperature of $\mathcal{L}_{OT}$) for OT-VAE on the test set of CelebA (Liu et al., 2015) and ZeroSpeech2019 (Dunbar et al., 2019). Results are illustrated in Table 5. We report MSE ($\times 10^{-3}$) for CelebA, MSE (dB$^2$) for ZeroSpeech2019 and perplexity (PPL) for both datasets. Two insights can be found from the table: first, with larger $\lambda$, PPL importantly boosts which validates the fact that optimal transport explicitly brings equipartition of the clusters; second, $\lambda = 1e - 3$ works decently across different modalities while the temperature needs to tuned according to different tasks.

## D    DETAILS OF EXPERIMENTS ON IMAGES

**Pre-processing**    Following SQ-VAE (Takida et al., 2022), all the images in CelebA (Liu et al., 2015) are center-cropped to $140 \times 140$ and resized to $64 \times 64$, which is a setting used in previous

Table 5: **Analysis of $\lambda$ and $\tau$ (regularization weight and initial temperature of $\mathcal{L}_{OT}$) for OT-VAE on the test set of CelebA (Liu et al., 2015) and ZeroSpeech2019 (Dunbar et al., 2019)**. We report MSE ($\times 10^{-3}$) for CelebA, MSE ($dB^2$) for ZeroSpeech2019 and perplexity (PPL) for both datasets. The codebook contains 512 codes and each code is with dimension 64. All the experiments are repeated *three* times.

| $\lambda$ | $\log \frac{1}{\tau}$ | CelebA | | ZeroSpeech2019 | |
|---|---|---|---|---|---|
| | | MSE ($\times 10^{-3}$) $\downarrow$ | PPL $\uparrow$ | MSE ($dB^2$) $\downarrow$ | PPL $\uparrow$ |
| 1e-4 | 0.1 | $0.96 \pm 0.0$ | $391.7 \pm 4.9$ | $32.62 \pm 0.97$ | $433.9 \pm 8.8$ |
| | 0.5 | $0.98 \pm 0.0$ | $373.3 \pm 2.7$ | $31.22 \pm 0.49$ | $435.8 \pm 4.6$ |
| | 1 | $0.98 \pm 0.0$ | $365.6 \pm 16.4$ | $32.65 \pm 1.06$ | $431.5 \pm 20.0$ |
| | 5 | $1.01 \pm 0.0$ | $377.1 \pm 22.4$ | $28.80 \pm 0.36$ | $438.8 \pm 3.6$ |
| | 10 | $0.99 \pm 0.0$ | $389.9 \pm 11.6$ | $31.37 \pm 0.79$ | $433.8 \pm 15.5$ |
| **1e-3** | 0.1 | $0.95 \pm 0.0$ | $425.4 \pm 4.9$ | $32.00 \pm 0.28$ | $448.2 \pm 2.1$ |
| | 0.5 | $0.96 \pm 0.0$ | $418.3 \pm 2.6$ | $30.76 \pm 0.41$ | $450.2 \pm 1.7$ |
| | **1** | $\mathbf{0.94 \pm 0.0}$ | $\mathbf{433.3 \pm 5.1}$ | $31.96 \pm 0.35$ | $452.9 \pm 2.0$ |
| | 5 | $0.95 \pm 0.0$ | $426.6 \pm 4.4$ | $\mathbf{28.73 \pm 0.29}$ | $\mathbf{446.2 \pm 6.5}$ |
| | 10 | $0.99 \pm 0.0$ | $424.3 \pm 2.8$ | $32.41 \pm 0.66$ | $443.5 \pm 3.0$ |
| 1e-2 | 0.1 | $0.96 \pm 0.0$ | $446.7 \pm 16.0$ | $32.15 \pm 0.09$ | $453.4 \pm 2.3$ |
| | 0.5 | $0.96 \pm 0.0$ | $450.2 \pm 3.7$ | $32.32 \pm 0.98$ | $445.2 \pm 4.1$ |
| | 1 | $0.96 \pm 0.0$ | $448.6 \pm 0.6$ | $30.79 \pm 0.43$ | $454.9 \pm 5.5$ |
| | 5 | $1.00 \pm 0.0$ | $450.9 \pm 3.2$ | $29.07 \pm 0.52$ | $449.5 \pm 1.7$ |
| | 10 | $1.00 \pm 0.0$ | $413.7 \pm 2.2$ | $30.61 \pm 0.83$ | $440.8 \pm 2.0$ |

work (Tolstikhin et al., 2018; Ghosh et al., 2020). For CelebAHQ-Mask (Liu et al., 2015), we center-crop the the segmentation maps to $128 \times 128$ and resized to $64 \times 64$ with the nearest neighbor interpolation.

**Architectures for reconstruction** We use the same architectures as SQ-VAE (Takida et al., 2022), which is adopted from the GitHub repository of DeepMind Sonnet [3] and composed of a ConvResNet-type encoder and decoder. These networks include convolutional layers, transpose convolutional layers, and ResBlocks, the downsampling rate from the image resolution to quantized layer is 4. Number of residual blocks are 6 for CelebA (Liu et al., 2015) and 2 for CelebAHQ-Mask (Liu et al., 2015).

**Model architectures for generation** As SQ-VAE did not release their implementation for generation, we reimplemented SQ-VAE using the same model architecture as for OT-VAE for a fair comparison. For both SQ-VAE (Takida et al., 2022) and OT-VAE, we train a standard GPT model (Vaswani et al., 2017; Radford et al., 2018) to generate images in an auto-regressive fashion. The GPT model has 12 layers, 12 heads, and feedforward dimension 768.

**Training details** Following SQ-VAE (Takida et al., 2022), the reconstruction loss $\mathcal{L}_{re}$ is the MSE loss for both datasets. We adopt the same optimization schema as SQ-VAE (Takida et al., 2022). we leverage Adam (Kingma & Ba, 2015) optimizer with inital learning rate 0.001, batch size 64, $\beta_1 = 0.9$ and $\beta_2 = 0.99$. The learning rate will be halved every 3 epochs if the validation loss is not improving. We train 200 epochs for CelebA and CelebAHQ-Mask. We validate on the same capacity of codebooks as SQ-VAE (Takida et al., 2022): the codebook contains 512 codes for CelebA and 64 codes for CelebAHQ-Mask, each code is with dimension 64. The training takes $\sim 2$ hours for CelebA using a single GPU NVIDIA V100-32G. The gumbel softmax decay is 1e-5 in all the datasets. The validation MSE and perplexity (PPL) on CelebA (Liu et al., 2015) during the training are illustrated in Figure 4a and 4b respectively, which shows the superiority of the proposed OT-VAE compared to SQ-VAE (Takida et al., 2022).

---

[3]Sonnet: https:sonnet/vqvae_example.ipynb

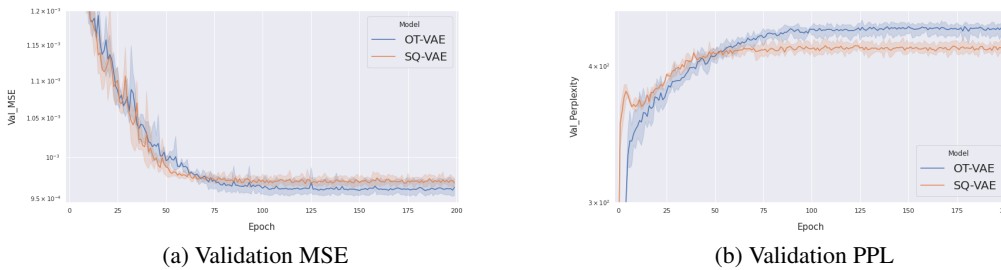

(a) Validation MSE          (b) Validation PPL

Figure 4: The validation MSE and perplexity (PPL) on CelebA (Liu et al., 2015) during the training. The propose OT-VAE obtains better reconstruction with higher perplexity. Experiments are repeated three times.

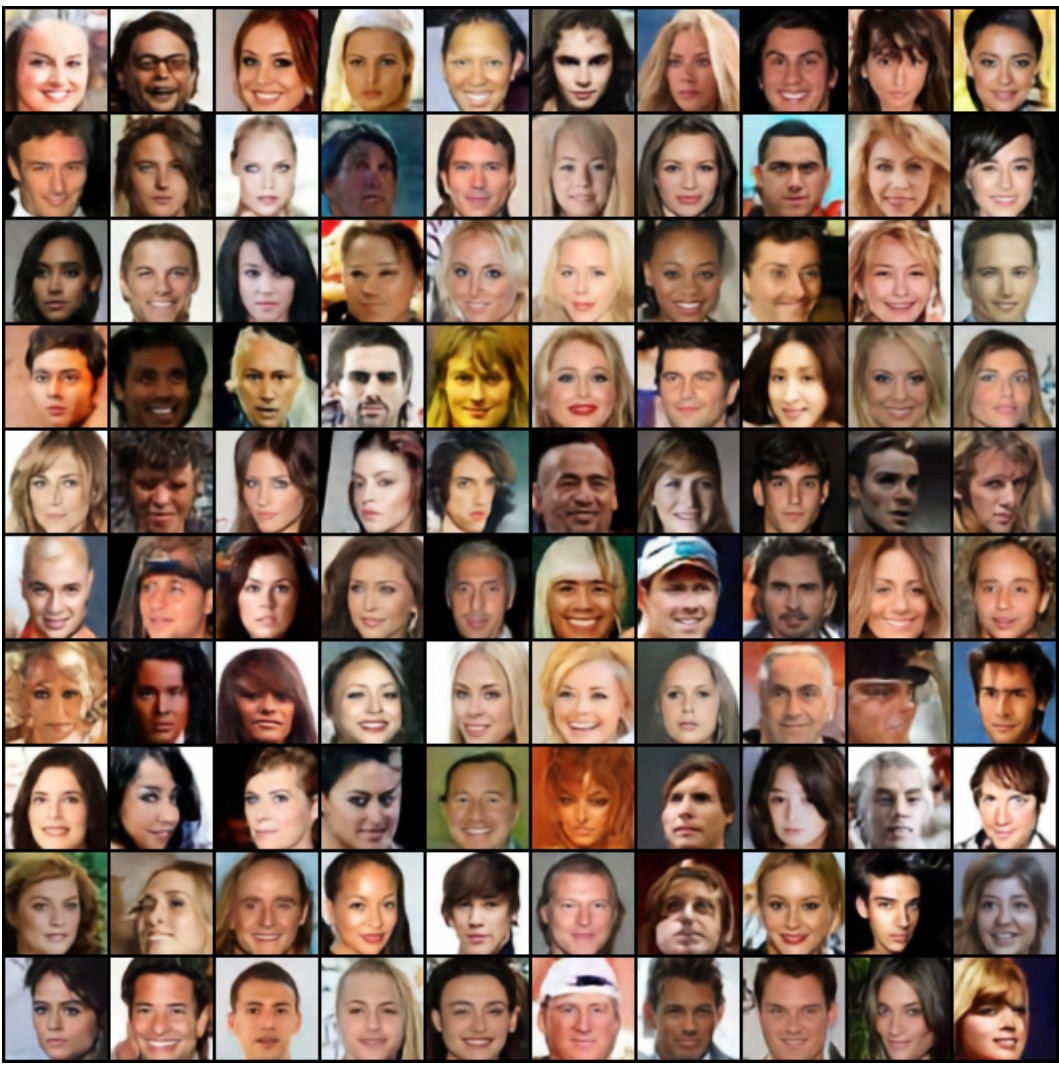

Figure 5: Generated images on CelebA (Liu et al., 2015) with the proposed OT-VAE.

For generation, an image is first encoded into a sequences of 256 tokens through the encoder of OT-VAE and SQ-VAE. As a standard strategy, a `SOS` token is padded to the beginning of each sequence. During training, we perform next token prediction with the teacher forcing. For generation. For generation, we start with the `SOS` token and use a simple multinomial sampling with the predicted

probability to generate the next token and repeat this step until the full length is reached. We train 300K iterations with AdamW (Loshchilov & Hutter, 2017) optimizer, and set the batch size to 16. We generate 10K images and use torch-fidelity [4] to compute the FID. We provide visual results in Figure 6.

## E    DETAILS OF EXPERIMENTS ON SPEECH DATA

**Pre-processing**    Following SQ-VAE (Takida et al., 2022), we apply the same pre-processing as van Niekerk et al. (2020) to extract 80-dimensional log-mel spectrogram features for all the speech data. We firstly scale the maximum amplitude of each audio signal to 0.999 and pre-emphasize the scaled audio signals with a first-order autoregressive filter ($y_t = x_t - 0.97x_{t-1}$, where $x_t$ and $y_t$ indicate the input and output of the filter at time $t$), then apply a 2048-point FFT with 25 ms Hann window, 10 ms frame shift, and frequency cutoffs in between 50 Hz and 8000 Hz. After that, we clip those bins that are 80.0 dB lower than the maximum and re-scale the log-mel spectrogram by dividing it by 80.

**Architecture**    The architecture for speech data is the same as SQ-VAE (Takida et al., 2022), which is with ConvResNet-type encoder/decoder, whereas the Conv2d for the image data is replaced by Conv1d to accommodate the structure of speech data. The downsampling rate from the mel spectrogram to the quantized layer is 2.

**Model architectures for generation**    Similar to image generation, we use the GPT architecture with 12 layers, 12 heads and 768 feedforward dimension.

**Training details**    Following SQ-VAE (Takida et al., 2022), we use Adam (Kingma & Ba, 2015) optimizer with inital learning rate 0.0004, batch size 256, $\beta_1 = 0.9$ and $\beta_2 = 0.99$. We train 50K iterations and pick the best model by validating our models on 200 samples, which are randomly selected in advance from the training set. The learning rate will be decayed with $\gamma = 0.5$ on 30,000 and 40,000 iterations. We have 512 codes with dimension 64 in the codebook, which is also the same as SQ-VAE (Takida et al., 2022). As suggested in Chorowski et al. (2019); Takida et al. (2022), we add a time-jitter regularization with a replacement probability of 0.5 during training, which is also used in SQ-VAE (Takida et al., 2022). The training takes $\sim 12$ hours using a single GPU NVIDIA V100-16G.

During training, we clip the waveform with a random start point to get a sequence of 100 tokens after the encoder of OT-VAE or SQ-VAE. Then we also performed next token generation with a **SOS token** concatenated at the beginning. We train 100k iterations with AdamW optimizer with the learning rate of $2e-4$, and the batch size of 32. For generation, we start with the **SOS token** and 15 warm-up tokens obtained from the encdoer, and autoregressively generated the next 85 tokens with top-k ($k = 5$) multinomial sampling. Then the 100 tokens in total can be fed into the decoder of OT/SQ-VAE to obtain the generated log-mel spectrograms.

To evaluate the quality of generated audio, we follow Bińkowski et al. (2019) to compute *Frechet distance* and the *Maximum Mean Discrepancy* between the generated data and the real data, which are also named *FDSD* and *KDSD* respectively in the reference paper. As suggested in Bińkowski et al. (2019), we obtained the features by feeding audio clips of *2s*-length into the pre-trained *Deep-Speech2* model (Amodei et al., 2016) from *NVIDIA OpenSeq2Seq* library (Kuchaiev et al., 2018) and averaged along the temporal dimension. Since the pre-trained model takes the raw waveforms as input, we also trained a *HiFi-GAN* model (Kong et al., 2020) on ZeroSpeech dataset to transform the mel-spectrogram into raw waveform. Due to the time limit, we used the simple V3 configuration, trained with batch size of 128 on 8 V-100 gpus and took the checkpoint at 180k to perform the waveform generation.

Considering that in the test set (*train_parallel_voice*), there are 159 out of 190 audio sequences that are longer than *2s*, we cropped 63 chunks with random start point for each audio, which resulted in a total of 10,017 audio clips for reasonable estimates.

---

[4] https://torch-fidelity.readthedocs.io/

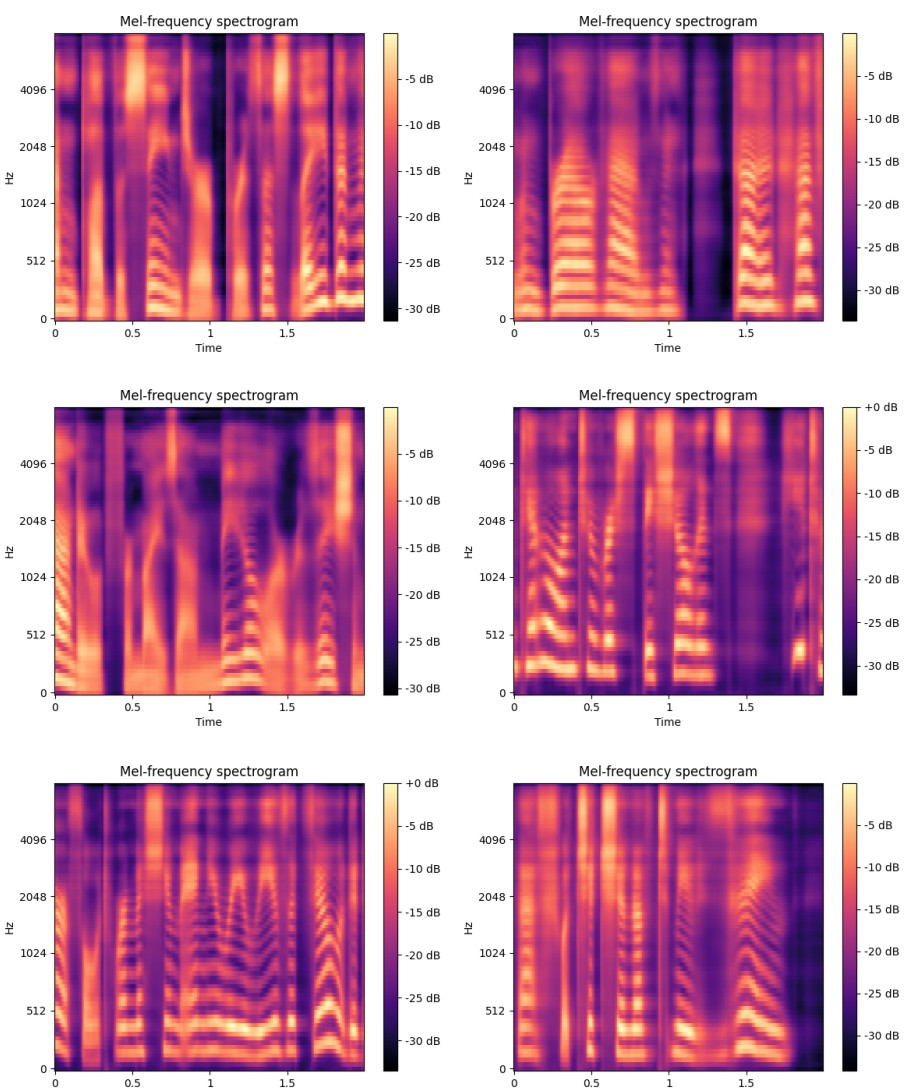

Figure 6: Generated audio spectrogram on ZeroSpeech2019 (Dunbar et al., 2019) with the proposed OT-VAE.

**Evaluation metrics**    Here we provide more details for the computation of Fréchet distance (i.e. FDSD) and Maximum Mean Discrepancy (i.e. KDSD). Given features from the target data $\boldsymbol{X} \in \mathbb{R}^{m \times d}$ and the features from the natural data $\boldsymbol{Y} \in \mathbb{R}^{n \times d}$, we have:

$$\text{FDSD}(\boldsymbol{X}, \boldsymbol{Y}) = ||\mu_{\boldsymbol{X}} - \mu_{\boldsymbol{Y}}||_2^2 + \text{Tr}\left(\Sigma_{\boldsymbol{X}} + \Sigma_{\boldsymbol{Y}} - 2(\Sigma_{\boldsymbol{X}}\Sigma_{\boldsymbol{Y}})^{1/2}\right) \tag{13}$$

$$\text{KDSD}(\boldsymbol{X}, \boldsymbol{Y}) = \frac{1}{m(m-1)} \sum_{1 \leq i,j \leq m, i \neq j} k(\boldsymbol{X}_i, \boldsymbol{X}_j) + \frac{1}{n(n-1)} \sum_{1 \leq i,j \leq n, i \neq j} k(\boldsymbol{Y}_i, \boldsymbol{Y}_j)$$
$$+ \sum_{i=1}^{m} \sum_{j=1}^{n} k(\boldsymbol{X}_i, \boldsymbol{Y}_j), \tag{14}$$

where $\mu_{\boldsymbol{X}}, \mu_{\boldsymbol{Y}}$ and $\Sigma_{\boldsymbol{X}}, \Sigma_{\boldsymbol{Y}}$ indicate the means and variances of $\boldsymbol{X}$ and $\boldsymbol{Y}$ respectively, whereas $k : \mathbb{R}^d \times \mathbb{R}^d \to \mathbb{R}$ is a positive definite kernel function. Follow Bińkowski et al. (2019; 2018), we

choose the polymial kernal:

$$k(\boldsymbol{x}, \boldsymbol{y}) = \left(\frac{1}{d}\boldsymbol{x}^T\boldsymbol{y} + 1\right)^3.$$

(15)

To evaluate the distribution of natural data, we split the test set in half and compute these two metrics between the two subsets.

## F  DETAILS OF EXPERIMENTS ON 3D DANCE GENERATION

**Our baseline (Li et al., 2022)**  Based on VQ-VAE, Li et al. (2022) is a two-stage framework for music to dance generation. In the first stage, it trains two VQ-VAEs to reconstruct 3D dance motions for upper and lower bodies separately. The training of VQ-VAE includes common heuristics such as: the commitment and embedding loss with stop-gradient operator ( Equation 1), exponential moving average (EMA) and codebook reset. In the second stage of training, a GPT is trained to generate 3D motion conditional on music sequences.

**Architecture**  We follow the official code of Li et al. (2022), which is released at `https://github.com/lisiyao21/Bailando`. The encoder-decoder architecture is composed of dilated 1D convolutions with kernel $4 \times 4$ and residual blocks. Convolutions with stride 2 are used in the encoder to reduce temporal resolution while transposed convolution with stride 2 are used in the decoder to increase the temporal resolution. The temporal downsampling rate is 8. Both codebooks for upper and lower bodies are with dimension $512 \times 512$. The GPT is composed a transformer with 12 layers and 12 heads. The embedding size in GPT is 768.

**Training details**  Denoted a dance motion as $X \in \mathbb{R}^{T \times (J \times 3)}$, where T is the temporal length and J = 24 is the total number of joints in SMPL (Loper et al., 2015), we learn two codebooks for upper and lower bodies separately. We use the same reconstruction loss as Li et al. (2022) consisting of the reconstruction between the input and the reconstructed motion as well as their the velocity $V(\cdot)$ and acceleration $A(\cdot)$, *i.e.* $\mathcal{L}_{re} = \mathcal{L}_1(X, G(Z)) + \mathcal{L}_1(V(X), V(G(Z))) + \mathcal{L}_1(A(X), A(G(Z)))$. We follow similar training schema as Li et al. (2022): we first train the motion without the root translation for 200K iteration then freeze the encoder, codebook and decoder to train an extra decoder for another 200K iteration to optimize the root velocity. We use Adam (Kingma & Ba, 2015) optimizer with learning rate 2e-4, batch size 128, T = 64, $\beta_1 = 0.9$, $\beta_2 = 0.99$.

We further train a Generative Pre-trained Transformer (GPT) for music to dance generation. We use music feature as Li et al. (2022). We encoder the motion into codebook index using our OT-VAE and train the GPT to predict the codebook index in an auto-regressive manner with cross-entropy loss. For inference, the motion is generated in an auto-regressive manner. The GPT is trained for 600K iteration with Adam (Kingma & Ba, 2015) optimizer. The learning rate is set to 2e-4 then decrease to 1e-5 at 400K iterations. We use batch size 32, T = 256, $\beta_1 = 0.9$, $\beta_2 = 0.99$.

