# OpenReview forum: "Learning Discrete Representation with Optimal Transport Quantized Autoencoders"
_ICLR.cc/2023/Conference — Submitted to ICLR 2023_

### Official Review · Reviewer_5Hey · 2022-10-24

**Confidence:** 4
**Correctness:** 3
**Technical Novelty And Significance:** 3
**Empirical Novelty And Significance:** 2
**Recommendation:** 5

**Clarity, Quality, Novelty And Reproducibility:**

The paper is clear, high quality, has novelty and includes a full pseudocode for reproducibility.

**Strength And Weaknesses:**

**Strengths**
- The paper is well written and the ideas are clearly explained.
- The experimental results show the method is able to achieve
  improvements in reconstruction accross multiple modalities (images,
  speech, 3D dance motion), and improvements in the task of 3D dance
  motion generation.

**Weaknesses**
- The experimental section is limited to small datasets. The most
  interesting application of VQ-VAEs is possibly  generation
  (learning the discrete latent prior). However, the impact of
  the proposed OT-VAE for generation is only demonstrated for one
  small (40 evaluation samples) 3D motion generation dataset, which in my opinion is
  too weak a signal for validating the improvements of the method for
  VQ-VAE-based generation.
- Unless I missed it, the comparison should also show the
  extra computation required to compute the $\Gamma$ matrix (9).
- I would suggest revising the general claim that the method "does not
  require heuristics" (abstract, page 2), by clearly specifying the
  kind of heuristic this is referring to.


**Summary Of The Paper:**

This paper proposes to regularize the codebook learning in a VQ-VAE
towards a uniform utilization of the codebook. The approach, termed
OT-VAE, uses a clustering equipartion objective in the latent space to
attain this goal. Experiments on small-scale datasets of three
different modalities show the approach improves the reconstruction
error and codebook utilization w.r.t. a VQ-VAE baseline, and improves
3d motion generation for one dataset.


**Summary Of The Review:**

The paper presents an interesting exploration into regularizing the
learned codebook of a VQ-VAE with an OT objective. The empirical
evidence for the impact of the proposed technique in learning the
discrete prior (e.g. for compression or generation), is very limited.
This is a weakness being these some of the main applications of
VQ-VAEs.

---

> ### Author Response · Authors · 2022-11-19
> **Response to Reviewer 5Hey**
>
> Thank you for your time and thoughtful comments. We have addressed the concerns below and in the revised manuscript. If you have any further questions, please let us know.
>
> > The experimental section is limited to small datasets. The most interesting application of VQ-VAEs is possibly generation (learning the discrete latent prior). However, the impact of the proposed OT-VAE for generation is only demonstrated for one small (40 evaluation samples) 3D motion generation dataset, which in my opinion is too weak a signal for validating the improvements of the method for VQ-VAE-based generation.
>
> The primary task we consider in this work is 3D motion generation, but existing datasets are generally small for this task, and AIST++ is the largest public dataset for dance generation to our knowledge. Our method outperforms existing state-of-the-art methods on this task. Based on your suggestion, we have also included generation results for image and speech data, as discussed in the __general comments__ above. We have also added more qualitative results in Figure 5 of the revised manuscript. While our method does not show significant improvement for image generation, please note that for discrete data distributions such as the CelebAHQ-Mask dataset, OT-VAE significantly outperforms existing methods (see Table 2), showing its potential for high-quality generation of complex categorical data.
>
> > Unless I missed it, the comparison should also show the extra computation required to compute the  matrix (9).
>
> Thank you for your suggestion. We have included computation time comparison for VQ-VAE and OT-VAE in Section 5.1 in the revised manuscript. Training one epoch on CelebA (of ~200k images) with optimal transport regularization takes about 102 seconds on a single GPU V-100, while it takes about 89 seconds without OT. The difference could be even small when using fewer iterations for Sinkhorn-Knopp algorithm. While computing the coupling matrix indeed requires extra computation time, please note that OT-VAE and VQ-VAE takes the same time for inference, as the OT coupling matrix is not needed.
>
> > I would suggest revising the general claim that the method "does not require heuristics" (abstract, page 2), by clearly specifying the kind of heuristic this is referring to.
>
> Thank you for your suggestion. We did specify the type of heuristics in the summarized list of our contribution at the end of page 2. Based on your suggestion, we have explicitly specified the kind of heuristic, namely stop-gradient, EMA, codebook reset, also for the abstract and introduction.

---

> > ### Comment · Reviewer_5Hey · 2022-12-12
> > **Thank you for addressing my comments.**
> >
> > Thank you for addressing my comments in the revised manuscript.
> > I appreciate the authors effort in training a GPT model to compare the OT and baseline encodings for generation. The marginal gains in generation obtained by the OT regularization seem low at the moment, and the FIDs reported for CelebA-64 seem very far from state of the art (which is <2, although typically using 50K samples). My score  would be higher if there was stronger evidence of a significant improvement in the trade-off between reconstruction quality and ease of modeling of the latent distribution.

---

> > > ### Author Response · Authors · 2022-12-14
> > > **Thank you for your feedback.**
> > >
> > > Thank you for your time and feedback. Regarding your comments, we would like to highlight the following points:
> > >
> > > - We would like to emphasize that the proposed OT regularization brings consistent improvement on several tasks with diverse data modalities, including image generation on CelebA (Table 1), segmentation mask generation on CelebAHQ-Mask (Table 2, categorical distribution), speech generation on ZeroSpeech2019 (Table 3) as well as motion generation on AIST++ (Table 4). In particular, on CelebAHQ-Mask and AIST++, OT-VAE shows substantial improvement compared to state-of-the-art methods.
> > >
> > > - We argue that the comparison between OT-VAE and state-of-the-art models on CelebA is unfair. Since our main focus is to solve the problem of codebook collapse in VQ-VAE, the most relevant work should be SQ-VAE (ICML'22). To this end, we use the same model architecture and training setup as SQ-VAE. Please note that the number of parameters is much smaller than other SOTA models for image generation on CelebA.
> > >
> > > - In addition to empirical results, we also provide a solid theoretical formulation under the framework of Wasserstein Auto-encoders, which paves the way for a better understanding of VQ-related models.
> > >
> > > We would be more than happy to discuss any further questions.

---

### Official Review · Reviewer_MP1B · 2022-10-25

**Confidence:** 4
**Clarity, Quality, Novelty And Reproducibility:** This paper is well written with suffi…
**Correctness:** 3
**Technical Novelty And Significance:** 3
**Empirical Novelty And Significance:** Not applicable
**Recommendation:** 6

**Strength And Weaknesses:**

#### **Strength**

  - The problem of Vector Quantization is timely and important, with increasing number of applications. The method is also well-motivated, as existing key challenges in VQ lie in low codebook usage.

 - The approach is well-represented. The authors make the tables and figures easy to follow and nicely-illustrated.

 - Extensive experiments across three different modalities are conducted. Notably, OT-VAE achieves state-of-the-art results on 3D dance generation.


#### **Weaknesses**

 - The authors stated that OT-VAE does not need to include other heuristics, such as stop-gradient, EMA, codebook reset. However, according to Table 5, the method is relatively sensitive towards regularization weight and initial temperature. Therefore, additional burdens on tuning these parameters exist.

 - The technique of L2 normalization has been adopted in ViT-VQGAN, which is not discussed here.

 - Quantitative improvements seem to be limited, especially for image modality, as there exists no qualitative comparisons except for 3D dance generation.

**Summary Of The Paper:**

 - As one key challenge of existing vector quantization methods comes from codebook collapse, this work proposes OT-VAE, which regularizes the quantization by explicitly assigning equal number of samples to each code.

 - The proposed method enforces the full utilization of the codebook while not requiring any heuristics.

 - Across three different data modalities, the authors empirically demonstrate that OT-VAE shows better reconstruction compared to other VQ-VAE variants.

**Summary Of The Review:**

As discussed in Strength and Weakness, I think this work is well-motivated. Meanwhile, with detailed figures and detailed explanations, the authors present OT-VAE clearly. Existing weak points about this work include: 1. some slightly-incorrect statement, such as the benefits over previous heuristics,  2. lack of some relevant discussions and comaprisons.

---

> ### Author Response · Authors · 2022-11-19
> **Response to Reviewer MP1B**
>
> Thank you for your time and thoughtful comments. We have addressed the concerns below and in the revised manuscript. If you have any further questions, please let us know.
>
> > according to Table 5, the method is relatively sensitive towards regularization weight and initial temperature. Therefore, additional burdens on tuning these parameters exist.
>
> Existing methods that use heuristics such as EMA and codebook reset require more sophisticated hyperparameter tuning, such as momentum and warm-up iterations for EMA, and the reset schedule for codebook reset, which typically result in much larger computational loads and requires significant engineering effort. Despite larger computational loads, these methods still do not guarantee no code collapse. In contrast, our work explicitly enforces the full usage of codebook by regularizing the quantization through optimal transport, which theoretically guarantees no collapse. In addition, our method only requires tuning a few hyperparameters, namely entropic regularization term $\lambda$ and initial temperature $\log\frac{1}{\tau}$, the choice of which only slightly affects the performance (see __Table 5__ in the Appendix). As a result, OT-VAE requires little engineering effort to achieve reasonably good performance.
>
> > The technique of L2 normalization has been adopted in ViT-VQGAN, which is not discussed here.
>
> Thank you for the suggestion. L2 normalization has indeed been used in ViT-VQGAN and we have added a brief discussion in Section 4.2.1 in the revised manuscript. Nevertheless, please note that our choice of using L2 normalization was primarily motivated by previous work in self-supervised contrastive learning, such as SwAV (Caron et al., 2021), rather than VQ related methods.
>
> > Quantitative improvements seem to be limited, especially for image modality, as there exists no qualitative comparisons except for 3D dance generation.
>
> Data compression is also one important application of VQ-VAE-like models and that's why we only showed reconstruction loss for image and speech data for a more direct comparison of the quantization quality. Based on your suggestion, we have also conducted experiments for comparing generation quality for images and speech as provided in the __general comments__ above. Visual results are also provided in Figure 5 of the revised manuscript.

---

### Official Review · Reviewer_cXiL · 2022-10-27

**Confidence:** 5
**Correctness:** 4
**Technical Novelty And Significance:** 3
**Empirical Novelty And Significance:** 3
**Recommendation:** 6

**Clarity, Quality, Novelty And Reproducibility:**

overall the manuscript is clear , the method part math equation should come with more explanation. The experiment part is clear and well conducted. The author uses an advanced and mathematically tight method to regularize code usage using a prior. But I am not sure if this advanced method perform better than other simpler approach following the same strategy.

**Strength And Weaknesses:**

Strength: the authors put the regularization of code usage into a detailed and solid mathematic framework of optimal transport and then use an Gumbel softmax max to sample the code in a one-hot manner. This strategy allows for a lot of flexibity and stochasticity of temperature applied. The experimental design is solid

weakness, not sure of the advanced framework is needed as the core of the problem is bias in code usage by regularizing the codebook usage. Therefore, I am not sure about the real novelty

**Summary Of The Paper:**

In this article the author seek to develop a method to solve the code collapsing problem which is common in VQ related methods. The nature of the proposed method is to regularize the code selection with a uniform prior so that different codes will be evenly used. The author conducted experiments in different settings including visual images, speech etc.

**Summary Of The Review:**

This manuscript present a sophisticated regularized version VQVAE and put the algorithm in OT framework. The manuscript is well written , the math is clean and the experiments are well organized. The largest concern I have is the novelty.

---

> ### Author Response · Authors · 2022-11-19
> **Response to Reviewer cXiL**
>
> Thank you for your time and thoughtful comments. We have addressed the concerns below and in the revised manuscript. If you have any further questions, please let us know.
>
> > not sure of the advanced framework is needed as the core of the problem is bias in code usage by regularizing the codebook usage.
>
> Code collapse is a well known issue in VQ related methods, and some strategies have been recently proposed to mitigate this issue, such as SQ-VAE (Takida et al., 2022). Other works use a couple of heuristics such as EMA and codebook reset, which requires significant engineering effort in practice. In contrast, our work explicitly enforces the full usage of codebook by regularizing the quantization through optimal transport, which theoretically guarantees no collapse. The benefits of our approach lie in the following aspects
>
> - Our approach provides a solid theoretical formulation under the framework of Wasserstein Auto-encoders, which paves the way for a better understanding of VQ related models. To our knowledge, our approach also demonstrates for the first time that techniques developed in self-supervised learning can be directly used for generative modeling.
> - Our approach does not require EMA, stop-gradient and codebook reset but only tuning a few hyperparameters, namely entropic regularization term $\lambda$ and initial temperature $\log\frac{1}{\tau}$, the choice of which only slightly affects the performance (see __Table 5__ in the Appendix). As a result, OT-VAE requires little engineering effort to achieve good performance.
> - For discrete data distributions such as the CelebAHQ-Mask dataset, OT-VAE significantly outperforms existing methods (see Table 2), showing its potential for high-quality generation of complex categorical data.
>
> > overall the manuscript is clear , the method part math equation should come with more explanation.
>
> We appreciate your suggestion and have updated the manuscript accordingly with more interpretation on the math equations in Section 4.1 and 4.2 of the revised manuscript (highlighted in red). If you have any further concrete suggestions, we would appreciate them.

---

> > ### Comment · Reviewer_cXiL · 2022-12-12
> > **Good reply**
> >
> > I am satisfied with the author's reply and like keep my score (there is no 7 this year...and I don't think it is strong enough to be 8)

---

> > > ### Author Response · Authors · 2022-12-14
> > > **Thank you for your feedback.**
> > >
> > > Thank you for your time and feedback!
> > > We would be more than happy to discuss any further questions.

---

### Author Response · Authors · 2022-11-19
**Rebuttal: general comments (1/2)**

Dear reviewers,

We would like to thank you for your time and effort in providing such helpful and detailed comments, which we believe have strengthened our paper considerably. We first thank you for your positive feedback:

- (Reviewer MP1B) The problem addressed in the paper is timely and important, with an increasing number of applications.
- (Reviewer cXiL, MP1B) The propose method is well-motivated, mathematically sound and has good flexibility.
- (Reviewer cXiL, MP1B, 5Hey) The paper is well written, the ideas are well explained and the approach is clearly presented.
- (Reviewer cXiL, MP1B) The experiments are well organized, solid, and extensive on datasets with three different modalities.

While we will address specific comments and concerns in the individual reviews (and also in the revised manuscript, where the changes can be seen in red), we want to address a general point here raised by several reviewers, namely additional experiments on generation. To this end, we have conducted additional experiments for image generation on CelbeA64 and speech generation on ZeroSpeech2019.

### Image generation on CelebA64

We have performed image generation with both OT-VAE and SQ-VAE on CelebA by training a GPT on the discrete latent priors for fair comparison, as SQ-VAE did not release their code for generation.

For both SQ-VAE and OT-VAE, we train a standard GPT model to generate the discrete code sequences representing the images, in an auto-regressive manner. The GPT has 12 layers, 12 heads, and FFN dimension 768. An image is first encoded into a sequences of 256 tokens through the encoder of OT-VAE and SQ-VAE. As a standard strategy, a __SOS token__ is padded to the beginning of each sequence. During training, we perform next token prediction with the teacher forcing. For generation, we start with the __SOS token__ and use a simple multinomial sampling with the predicted probability to generate the next token and repeat this step until the full length is reached. We train 300K iterations with AdamW optimizer and set the batch size to 16. We generate 10K images and use [torch-fidelity]([torch-fidelity.readthedocs.io](https://torch-fidelity.readthedocs.io/)) to compute FID. The results are given in the table below

| Method                      | FID  |
| --------------------------- | ---- |
| SQ-VAE                      | 28.1 |
| SQ-VAE (Our implementation) | 20.8 |
| **OT-VAE**                      | **20.0**|

We observe that i) our reproduction is much better than SQ-VAE, which might be the reason of using GPT instead of PixelCNN; ii) under our fair setting, OT-VAE perform slightly better than SQ-VAE. Qualitative results are provided in Figure 5 of the Appendix in the revised manuscript.

---

> ### Author Response · Authors · 2022-11-19
> **Rebuttal: general comments (2/2)**
>
> ### Speech generation on ZeroSpeech2019
>
> We also conducted speech generation on ZeroSpeech2019. Similar to aformentioned image generation experiments, we trained a GPT on the discrete latent priors. During training, we clip the waveform with a random start point to get a sequence of 100 tokens after the encoder of OT-VAE or SQ-VAE. Then we also performed next token generation with a **SOS token** concatenated at the beginning. We train 100k iterations with AdamW optimizer with the learning rate of 2e-4, and the batch size of 32. For generation, we start with the **SOS token** and 15 warm-up tokens obtained from the encdoer, and autoregressively generated the next 85 tokens with top-k (k=5) multinomial sampling. Then the 100 tokens in total can be fed into the decoder of OT/SQ-VAE to obtain the generated log-mel spectrograms.
>
> To evaluate the quality of generated audio, we follow [1] to compute *Frechet distance* and the *Maximum Mean Discrepancy* between the generated data and the real data, which are also named *FDSD* and *KDSD* respectively in the reference paper. As suggested in [1], we obtained the features by feeding audio clips of *2s*-length into the pre-trained *DeepSpeech2* model [2] from *NVIDIA OpenSeq2Seq* library [3] and averaged along the temporal dimension. Since the pre-trained model takes the raw waveforms as input, we also trained a *HiFi-GAN* model [4] on ZeroSpeech dataset to transform the mel-spectrogram into raw waveform. Due to the time limit, we used the simple V3 configuration, trained with batch size of 128 on 8 V-100 gpus and took the checkpoint at 180k to perform the waveform generation.
>
> Considering that in the test set (*train_parallel_voice*), there are 159 out of 190 audio sequences that are longer than *2s*, we cropped 63 chunks with random start point for each audio, which resulted in a total of 10,017 audio clips for reasonable estimates.
>
>
> | Method                      | FDSD  | KDSD (x10e-3) |
> | --------------------------- | ----- | ----- |
> | Natural speech              | 5.82  | 3.03  |
> | SQ-VAE (Resynthesis)        | 8.95 +/- 1.11  | 14.55 +/- 1.98 |
> | **OT-VAE (Resynthesis)**    | **7.29 +/- 0.44**  | **11.61 +/- 0.78** |
> | SQ-VAE (Generation)         | 10.90 | 15.09 |
> | **OT-VAE (Generation)**     | **10.60** | **14.55** |
>
> We report the *FDSD* and *KDSD* scores of the generated audio from OT-VAE and SQ-VAE. We also report the scores from the *natural speech* by splitting the test set in half, and the scores from *Resynthesis* by directly feeding the speech into the AutoEncoder. We obersed that OT-VAE still perform better than SQ-VAE both in resynthesis and in generation.
>
>
> [1] Bińkowski, Mikołaj, et al. "High fidelity speech synthesis with adversarial networks.", ICLR 2020
> [2] Amodei, Dario, et al. "Deep speech 2: End-to-end speech recognition in english and mandarin." ICML, 2016.
> [3] Oleksii Kuchaiev, et al. "OpenSeq2Seq: Extensible toolkit for distributed and mixed precision training of sequence-to-sequence models."" In NLP-OSS, 2018.
> [4] Kong, Jungil, et al. "Hifi-gan: Generative adversarial networks for efficient and high fidelity speech synthesis." NeurIPS, 2020.

---

### Decision · Program_Chairs · 2023-01-20

**Decision:**

Reject

**Justification For Why Not Higher Score:**

Reviewers recommended showing better experimentation of the method and to compare it with other regularizers proposed in the literature for quantized VAE on large scale problems.

**Justification For Why Not Lower Score:**

N/A

**Metareview: Summary, Strengths And Weaknesses:**

**Summary**

VQ-VAE are known for their difficulty in training due to collapse of the codebook. The paper casts VQ-VAE in the Wasserstein autoencoder setting , and proposes to regularize the distribution of the codebook with the uniform distribution using an OT cost. Authors use Gumbel softmax to sample from the codebook to avoid the hard assignment. The paper is well written and easy to follow.

During rebuttal authors provided experiments in image and speech generation, and they had before experiments on 3D motion.

* Reviewer cXiL main concern was that the novelty

* Reviewer cXiL  main concern is about the usage of a heuristic (L_2 reg.) that was proposed in a previous work and the lack of substantial improvements

* Reviewer 5Hey) concern is about the impact of the work and the small dataset used for evaluating it.


**Summary Of Ac-Reviewer Meeting:**

After the discussion with the reviewers the following points were raised by the reviewers:
* good usage of OT to regularize the  codebook to prevent its collapse
* Limited novelty
* After the rebuttal authors added new experiments on speech and CelebA nevertheless the reviewer thought it is still lacking convincing  large scale experiments or substantial improvement on state of the art on celebA for e.g